# SHAPE2GCODE: DIRECT G-CODE GENERATION FROM 3D SHAPE DATA FOR AUTOMATED MANUFACTURING

## ABSTRACT

Modern manufacturing relies on Computer Numerical Control (CNC) machines, which execute machining operations using G-code, a programming language that defines tool movements, cutting paths, and machining parameters. Despite advancements in automation, generating G-code still requires significant human intervention and reliance on Computer-Aided Manufacturing (CAM) tools. To address these challenges, we propose Shape2Gcode, an end-to-end framework that directly generates optimized G-code from 3D shape data. Our approach leverages reinforcement learning to optimize key machining parameters, including tool radius, milling depth, and toolpath strategies. Additionally, Shape2Gcode incorporates a tool orientation selection module to determine optimal rotation matrices, enhancing the flexibility and precision of the machining. We evaluate Shape2Gcode on CNC manufacturing tasks using the ABC and ShapeNet datasets, comparing its performance against existing CAD reconstitution and CNC automation methods. Experimental results demonstrate that Shape2Gcode outperforms conventional approaches in reconstruction accuracy, significantly reducing the need for manual intervention. By optimizing G-code generation and minimizing manual adjustments, Shape2Gcode improves CNC manufacturing efficiency, lowers costs, and enables more automated machining workflows.

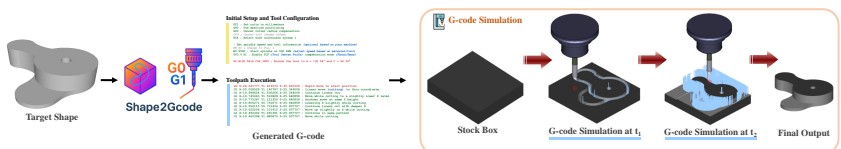

Figure 1: **Overview of our proposed pipeline.**

## 1 INTRODUCTION

Manufacturing has undergone a significant transformation with the advent of Computer Numerical Control (CNC) machining, which automates tool movement through G-code instructions. Traditional Computer-Aided Manufacturing (CAM) workflows rely on Computer-Aided Design (CAD) models to generate toolpaths before converting them into G-code. This multi-stage process is widely used in CNC machining, but optimizing G-code remains a manual and time-intensive task, as experts must fine-tune machining parameters such as tool radius, cutting depth, and toolpath strategies to achieve high-quality manufacturing outcomes. Expert involvement restricts automation and extends production time. A system that automatically generates optimized G-code directly from 3D shapes, without CAD conversion, could significantly boost manufacturing efficiency.

Recent deep learning-based approaches, such as CNC-Net, aim to infer machining operations directly from 3D models without relying on labeled datasets. While CNC-Net represents a significant advancement, it has several critical limitations. First, it does not account for real-world machining constraints, making it difficult to apply in practical CNC manufacturing. For instance, it fails to model layer-by-layer material removal, which is essential for milling operations that must avoid direct vertical tool movements. Additionally, CNC-Net's large decision space for tool parameters results in unstable and imprecise toolpaths, limiting its effectiveness in high-precision machining applications.

Figure 2: **The framework of Shape2Gcode.** The large box depicts the training episodes of Shape2Gcode, accompanied by a section presenting the reward and action space components.

To overcome these challenges, we introduce Shape2Gcode, the first method that directly generates G-code from 3D shape data without requiring CAD conversion. Our approach includes a tool orientation selection module that determines a minimal yet sufficient set of orientations to enusre complete surface coverage. Additionally, Shape2Gcode leverages reinforcement learning to autonomously learn machining strategies and select the best toolpath parameters, including tool radius, layer height, and tool strategy. Unlike previous methods, Shape2Gcode reduces computational complexity by focusing on critical machining parameters while maintaining precision. It constructs toolpaths using conventional machining patterns (e.g., Contour, Zigzag, Periphery, and Spiral), ensuring compatibility with standard CAM practices. Compared to CNC-Net, our method significantly reduces the decision space, leading to more stable and precise toolpaths. We evaluate Shape2Gcode on benchmark datasets and real-world G-code simulators, demonstrating that it outperforms existing deep learning approaches in accuracy and efficiency. By automating G-code optimization, Shape2Gcode enhances CNC manufacturing by making machining faster, more precise, and less dependent on human expertise, ultimately improving modern manufacturing workflows.

We summarize our main contributions as follows:

- **End-to-End G-code Generation:** Shape2Gcode directly generates G-code from 3D shape data, eliminating the need for CAD conversion and manual CAM workflows.
- **Reinforcement Learning for Machining Optimization:** Our model autonomously optimizes machining parameters, including tool radius, layer height, and toolpath strategy, ensuring efficiency and precision.
- **Real-World Compatibility:** Shape2Gcode is the first approach to generate G-code compatible with real-world CNC simulators.

## 2 RELATED WORKS

### 2.1 REVERSE ENGINEERING FOR 3D SHAPES

Reverse engineering in 3D manufacturing digitizes and reconstructs objects for replication, modification, and inspection, aiming to recover their geometric and structural features for fabrication analysis. Deep learning has enabled primitive-based modeling, approximating shapes with cubes Tulsiani et al. (2017); Zou et al. (2017); Niu et al. (2018), ellipsoids Genova et al. (2019), or deformable primitives Deng et al. (2020); Yavartanoo et al. (2021); Paschalidou et al. (2021); Huang et al. (2023), often using constructive solid geometry (CSG) Laidlaw et al. (1986); Foley et al. (1996). Reinforcement learning has also been applied to sequential primitive assembly Sharma et al. (2018); Du et al. (2018); Chung et al. (2021). Other approaches detect and fit primitives within point clouds Li et al. (2019); Sharma et al. (2020) or infer CSG programs Sharma et al. (2018); Du et al. (2018). Recent works compress CAD models via CSG operations without ground-truth assemblies Kania et al. (2020); Ren et al. (2021; 2022); Yu et al. (2022); Li et al. (2023). However, most methods focus on static reconstruction and overlook the sequential material removal essential for CNC machining, motivating the development of frameworks that capture its stepwise nature.

## 2.2 CNC Machining for Reverse Engineering

CNC machining is widely used for material removal and part reproduction in reverse engineering. Although CNC machines follow G-code precisely, generating optimized code remains manual and labor-intensive, requiring expert tuning of parameters and toolpaths. To address this, machine learning-based Computer-Aided Process Planning (CAPP) has been explored, using techniques such as particle swarm optimization (PSO) and support vector machines (SVM) Hsieh & Chu (2013); Dittrich et al. (2019). However, these approaches depend on pre-labeled CAD models and shows limited generalization ability across diverse datasets. Route planning methods Balic & Korosec (2002); Kukreja & Pande (2023) also generate optimized toolpaths, but require extensive preparation of CAD-toolpath pairs, making training time-consuming and limiting generalization. Given these limitations, research on automatically searching and learning CNC operations in a sequential manner remains limited. CNC-Net Yavatanoo et al. (2024) proposed a self-supervised framework that learns operations from 3D models in an unsupervised manner. Despite this progress, it does not fully capture machining constraints and often produces infeasible toolpaths due to its large decision space and indirect feedback. To overcome these challenges, we propose Shape2Gcode, an RL-based framework that directly generates optimized G-code from 3D shapes without CAD conversion, autonomously learning machining strategies for more efficient, stable, and precise operations.

## 3 Method

In this section, we introduce Shape2Gcode, a novel framework that translates 3D shape representations into optimized G-code for CNC machining. Instead of relying on traditional CAD-based workflows, our method directly learns to generate efficient machining plans from raw 3D shape inputs. Using a Deep Q-Network (DQN), it dynamically selects the optimal toolpath strategy, tool radius, and layer height, enabling a data-driven and adaptive approach to toolpath generation. By eliminating manual parameter tuning, Shape2Gcode improves efficiency and precision in CNC machining.

As shown in Figure 2, Shape2Gcode formulates G-code generation as an RL task, where an agent optimizes machining parameters by interacting with the environment to carve material from the stock material that encloses the target shape. The entire machining process is modeled as an episode, consisting of a sequence of decision-making steps, each contributing to the final toolpath generation. The pipeline begins with a 3D shape representation $\mathcal{S}$ of the target object. An angle selection module first determines valid machining orientations $V = \{v_s\}_{s=1}^n$ to ensure full surface coverage. Each selected angle provides a different view of the object, ensuring that critical features are not occluded during machining. For each view direction $v_s$, a visibility-based reconstruction step refines the input shape as $\mathcal{S}_s$ by ensuring that all visible regions align with the target shape, while occluded regions are filled. This guarantees that the machining operations are applied to an optimal, manufacturable representation. Once the shape $\mathcal{S}_s$ is reconstructed for a given machining angle $v_s$, an encoder $\mathcal{E}$ extracts a compact latent representation of the geometry:

$$\mathbf{z}_s = \mathcal{E}(\mathcal{S}_s), \tag{1}$$

where $\mathbf{z}_s \in \mathbb{R}^d$ captures essential geometric and machining features. Using this learned features, a DQN determines the optimal machining parameters, including tool radius $r_s^{\text{tool}}$, layer height $h_s^{\text{layer}}$, and toolpath strategy $s_s^{\text{strategy}}$, by selecting an action $a_s$ in the actions space $\mathcal{A}$ that maximizes the expected reward:

$$a_s = \{r_s^{\text{tool}}, h_s^{\text{layer}}, s_s^{\text{strategy}}\} \in \mathcal{A}. \tag{2}$$

The generated parameters are then translated into executable G-code as machine-readable instructions through a G-code generator. At the end of the episode, the complete G-code sequence, corresponding to all selected machining angles, is then sent to a CNC machine simulator for validation. The agent receives intermediate rewards at each step, as well as a final reward at the end of the episode, allowing the learned policy to optimize both the step-wise machining behavior and the overall outcome. The DQN policy is updated using this feedback, refining its decision-making for future episodes. By integrating reinforcement learning with G-code generation, Shape2Gcode learns to dynamically optimize toolpaths, ensuring efficient, precise, and automated CNC machining across various 3D shapes. In the following sections, we provide a detailed discussion of the tool orientation selection module, visibility-based reconstruction, G-code generator, and reward calculation, explaining how each component contributes to the overall machining process.

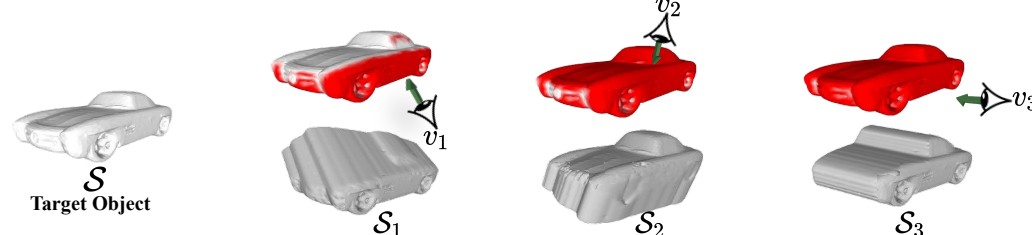

Figure 3: **Tool orientation selection process and visibility-based reconstructions. Top:** Accumulated coverage with added orientations. **Bottom:** Visibility-based reconstructions per orientation.

## 3.1 TOOL ORIENTATION SELECTION

The goal of the tool orientation selection module is to identify a minimal set of tool orientations that enables complete and efficient machining of a 3D shape. Since a shape cannot be fully machined from a single direction due to occlusions, it is necessary to select a set of orientations $V = {v_s}_{s=1}^n$ that achieves full surface coverage with as few orientations as possible. A surface point $p$ is considered visible from a candidate orientation $v$ if the cutting tool can reach $p$ directly, without obstruction. Mathematically, visibility is determined by casting a ray from $p$ along $-v$ and checking for intersections with the object surface; if the ray hits another surface point before exiting, $p$ is occluded in direction $v$. To select optimal tool directions, we use an iterative greedy approach: at each step $i$, the next orientation $v_i$ is chosen to maximize the number of uncovered visible points:

$$v_i = \arg\max_v \sum_{p \in P_{\text{uncovered}}} \mathbf{1}(\text{Vis}(p, v)), \tag{3}$$

where $P_{\text{uncovered}}$ is the set of surface points not yet covered, and $\mathbf{1}(\text{Vis}(p, v))$ is 1 if $p$ is visible from $v$, and 0 otherwise. The process stops once sufficient coverage is achieved, yielding the minimal set $V = \{v_0, v_1, \ldots, v_n\}$ of tool orientations.

## 3.2 VISIBILITY-BASED RECONSTRUCTION

The visibility-based reconstruction step processes the input shape $\mathcal{S}$ to generate a manufacturable representation $\mathcal{S}_s$, ensuring that all visible regions align with $\mathcal{S}$ while occluded regions, corresponding to the view direction $v_s$, are filled as shown in Figure 3. This ensures that machining operations are applied to a manufacturable and structurally valid representation, preserving the integrity of the object while facilitating effective tool movements. To achieve this, the reconstruction process employs a surface voxelization approach, converting the visible portions of $\mathcal{S}$ into a discrete volumetric grid. The occluded regions are filled to form a watertight representation. The reconstructed shape is then refined using marching cubes to generate a smooth mesh representation, followed by Laplacian smoothing to enhance surface continuity.

## 3.3 G-CODE GENERATION

The G-code generation module converts the generated action including the optimal machining parameters into executable CNC commands, ensuring that the CNC machine correctly interprets the learned machining strategy. This step bridges the reinforcement learning-based toolpath generation and the actual machining process by encoding the tool's movements in a standardized format. The machining process is carried out layer by layer along the depth direction, ensuring systematic material removal with the depth level of $h_s^{\text{layer}}$. At each layer $l$, the contour toolpath is first applied to separate the object's main geometry from the surrounding material. However, since the detached material remains connected to deeper layers, additional toolpath strategies e.g., zigzag, periphery, and spiral, are employed to efficiently remove the remaining stock. The reinforcement learning model selects the most efficient strategy for each layer. To maintain machining precision and prevent unintended collisions, the toolpath is dynamically adjusted. In each layer, the tool retracts to a starting position $(x_{s,\text{st}}, y_{s,\text{st}}, z_{s,\text{st}})$ executing:

$$G_{s,\text{st}} = \texttt{G0}\ X x_{s,\text{st}}\ Y y_{s,\text{st}}\ Z z_{s,\text{st}}, \tag{4}$$

where `G0` refers to the transition to the starting position. This ensures that the tool moves safely without damaging the target shape. In the following subsections, we provide a detailed explanation of each toolpath strategy, including contour, zigzag, periphery, and spiral.

### 3.3.1 CONTOUR TOOLPATH

The contour toolpath is designed to refine the object's surface by closely following its geometry. The process begins with the visibility-reconstructed shape $\mathcal{S}_s$, where a signed distance function is calculated as $\text{SDF}_s(p)$ for each point $p \in P_l$ in the layer $l$ to its surface. The contour toolpath points $P_{s,l}^{\text{surf}}$ are determined by selecting points where the SDF value is approximately equal to the tool radius, ensuring that the tool follows the object's surface while maintaining an appropriate clearance:

$$P_{s,l}^{\text{surf}} = \{p \in P_l; \left|\text{SDF}_s(p) - r_s^{\text{tool}}\right| < \epsilon\}, \tag{5}$$

where $\epsilon$ is a small tolerance to account for numerical precision. The sorting process follows a nearest-neighbor approach, starting from a starting point and iteratively selecting the closest unprocessed point until all points are visited and the G-code is updated:

$$G_{s,l,i} = \text{G1 } X x_{s,l,i} \, Y y_{s,l,i} \, Z z_{s,l,i}, \tag{6}$$

where $(x_{s,l,i}, y_{s,l,i}, z_{s,l,i})$ is the location of $p_{s,l,i} \in P_{s,l}^{\text{surf}}$ in $\mathbb{R}^3$ and `G1` indicates carving motion.

### 3.3.2 ZIGZAG TOOLPATH

The zigzag toolpath is designed for efficient material removal by systematically traversing the remaining stock within a machining layer. At each depth level $l$, after the contour toolpath has separated the object from the surrounding material, the zigzag strategy ensures that unremoved regions are cleared. The toolpath follows a structured back-and-forth motion, covering the machining layer in a sweeping manner. The traversal path is determined by selecting all points $P_{s,l}^{\text{out}}$ where the signed distance function value is greater than the tool radius:

$$P_{s,l}^{\text{out}} = \{p \in P_l; \text{SDF}_s(p) > r_s^{\text{tool}}\}. \tag{7}$$

The path begins at the starting point within the current viewpoint and progresses horizontally to the right. Once the tool reaches the boundary, it moves one step upward and then traverses back to the left. This alternating movement continues until all designated points in the layer have been processed the G-code is updated:

$$G_{s,l,i} = \text{G1 } X x_{s,l,i} \, Y y_{s,l,i} \, Z z_{s,l,i}, \tag{8}$$

where $(x_{s,l,i}, y_{s,l,i}, z_{s,l,i})$ is the location of a point $p_{s,l,i} \in P_{s,l}^{\text{out}}$ in the 3D space $\mathbb{R}^3$.

### 3.3.3 PERIPHERY TOOLPATH

Similar to the zigzag toolpath is designed for efficient material removal by traversing the remaining stock within a machining layer. However, unlike the zigzag strategy, where points are visited in a structured back-and-forth motion, the periphery toolpath orders the points in a counterclockwise sequence around the starting point in the layer $l$ and the G-code is updated as equation 8.

### 3.3.4 SPIRAL TOOLPATH

Similar to the zigzag and periphery toolpaths, the spiral toolpath is designed for efficient material removal by traversing the remaining stock within a machining layer. Instead of following a counter-clockwise sequence, the tool moves along a continuous spiral trajectory outward from the starting point, ensuring smooth and uniform material removal, and the G-code is updated as equation 8.

### 3.4 REWARD CALCULATION

The core decision-making process of Shape2Gcode is powered by a DQN, which learns a policy to select machining parameters by maximizing a reward function. To assess the efficacy of the generated parameters and precise object production, we simulate the CNC cutting process in a voxelized environment. The target shape $\mathcal{S}$ and the shape at each step $s$ are represented as a binary grids:

$$V_t(i,j,k) = \begin{cases} 1 & \text{inside}, \\ 0 & \text{outside}, \end{cases} \quad V_s(i,j,k) = \begin{cases} 1 & \text{if material is present}, \\ 0 & \text{otherwise}, \end{cases} \quad (i,j,k) \in \mathbb{Z}^3, \quad (9)$$

where initially all voxels $V_{s=0}$ are set as one. During simulation, the toolpath modifies the values, , setting the removed voxels by the tool to $0$. We define the overall reward as a combination of preservation of the object structure, precision of the final shape, and machining efficiency.

### 3.4.1 PRESERVATION ACCURACY

At each step $s$, we ensure that voxels inside the target shape are not cut in $V_s$ and retain values of 1:

$$\mathcal{R}_s = \frac{\sum_{(i,j,k) \in V_t^{in}} V_s(i,j,k)}{|V_t^{in}|}, \tag{10}$$

where $|V_t^{in}|$ is total number of voxels inside the target shape. This term penalizes any overcutting into regions that should be preserved.

### 3.4.2 FINAL SHAPE ACCURACY

To evaluate the accuracy of the final shape, we compute the IoU between the the $V_{s=n}$ and $V_t$:

$$\mathcal{R}_f = \text{IoU}(V_{s=n}, V_t). \tag{11}$$

This terms ensures that the target shape is precisely reproduced.

### 3.4.3 MATERIAL REMOVAL EFFICIENCY

We also consider machining efficiency which is quantified by the material removal rate:

$$\mathcal{R}_{\text{MRR}} = \frac{|1 - V_{s=n}|}{t_{\text{cut}}}, \tag{12}$$

where $|1 - V_{s=n}|$ is the total number of voxels removed, and $t_{\text{cut}}$ is the execution time.

### 3.4.4 TOTAL REWARD FUNCTION

The total reward function is a linear combination of all defined reward terms:

$$\mathcal{R}_{\text{total}} = \lambda_s \sum_{s=1}^{s=n} \mathcal{R}_s + \lambda_f \mathcal{R}_f + \lambda_{\text{MRR}} \mathcal{R}_{\text{MRR}}, \tag{13}$$

This reward structure enables the learning agent to generate high-quality and efficient toolpaths that minimize damage to the object while maximizing alignment with the final desired geometry.

### 3.5 TOOLPATH ACTION SPACE AND NETWORK ARCHITECTURE

The action space comprises tool radius ($0.001$–$0.01m$), layer height ($1/64$–$1/512m$), and toolpath strategy. Among toolpath startegies, contour is always chosen to preserve surfaces, while one of the others is selected for material removal. DGCNN Wang et al. (2019) extracts shape features, and a DQN selects machining actions. Further implementation details are provided in the Appendix A.1.

## 4 EXPERIMENTS

### 4.1 DATASETS

**ABC Dataset.** The ABC datasetKoch et al. (2019) contains one million 3D CAD models, primarily designed for manufacturing applications. It serves as a valuable resource for developing geometric deep-learning methods. As in CNC-Net, we pre-train our model using 5,000 normalized single-part CAD objects. Due to the computational cost of fine-tuning on individual shapes, we randomly sample 50 shapes from a set of 1,000 test samples for fine-tuning and evaluation.

**ShapeNet Dataset.** To ensure a broader generalization, we also utilize ShapeNet Core (V1)Chang et al. (2015), a dataset containing diverse 3D objects. We use watertight shapes obtained from ONetMescheder et al. (2019) for training and evaluation. Consistent with CAPRI-Net, we pre-train our model on 35,000 shapes across 13 categories and randomly select 10 shapes per category from the test set for fine-tuning and evaluation.

## 4.2 Evaluation Protocol

For quantitative evaluation, we execute the generated G-code in a voxel-based simulator on a $256^3$ grid aligned to the normalized cube $[-0.5, 0.5]^3$, where each motion segment removes voxels of the stock intersected by the swept tool volume (see Sec. A.2 for details). After simulation, the residual stock is converted into a mesh using Marching Cubes at the same resolution, without further post-processing.

We use public code/weights when available, otherwise we retrain on the same split. All baselines are run under their per-sample fine-tuning protocol, which is common in manufacturing where achieving high-fidelity results is prioritized. For completeness, we also conduct experiments without fine-tuning on CNC-Net and our method. Non-mesh outputs (CAD, implicit fields, toolpaths) are converted to meshes with the same Marching Cubes configuration to ensure parity, and all meshes are evaluated with the unified metric pipeline in Section 4.3.

## 4.3 Evaluation Metrics

**Volume-Based Metrics.** We use Intersection over Union (IoU)Yu et al. (2021) and F1-scoreRen et al. (2022) to assess the accuracy of reconstructed shapes. As in previous works, we voxelize the bounding box $[-0.5, 0.5]^3 \subset \mathbb{R}^3$ into $256^3$ voxels and evaluate their occupancies against the reconstructed meshes.

**Surface-Based Metrics.** To measure geometric accuracy, we use symmetric Chamfer Distance (CD)Mittal et al. (2021) and Normal Consistency (NC)Chen et al. (2020). Following previous works Yu et al. (2022); Yavatanoo et al. (2024), we uniformly sample 8,000 points on the surface of each object, with all CD values scaled by 1,000 for consistency.

## 4.4 Quantitative and Qualitative Results

| Method | Finetuning | ABC | | | | ShapeNet | | | |
|---|---|---|---|---|---|---|---|---|---|
| | | IoU↑ | F1↑ | CD↓ | NC↑ | IoU↑ | F1↑ | CD↓ | NC↑ |
| CSG-Stump Ren et al. (2021) | ✓ | 0.787 | 0.879 | 0.428 | 0.884 | 0.697 | 0.827 | 0.521 | 0.866 |
| ExtrudeNet Ren et al. (2022) | ✓ | 0.769 | 0.875 | 0.505 | 0.871 | 0.607 | 0.773 | 0.918 | 0.844 |
| CAPRI-Net Yu et al. (2022) | ✓ | 0.768 | 0.866 | **0.312** | 0.914 | 0.700 | 0.824 | **0.447** | **0.895** |
| SECAD-Net Li et al. (2023) | ✓ | 0.776 | 0.867 | 0.398 | 0.900 | 0.650 | 0.784 | 2.405 | 0.852 |
| CNC-Net Yavatanoo et al. (2024) | ✗ | 0.780 | 0.889 | 1.127 | 0.864 | 0.698 | 0.817 | 2.154 | 0.844 |
| CNC-Net Yavatanoo et al. (2024) | ✓ | 0.824 | 0.901 | 0.509 | 0.893 | 0.740 | 0.850 | 1.562 | 0.863 |
| Ours | ✗ | 0.833 | 0.902 | 0.493 | 0.914 | 0.766 | 0.858 | 0.678 | 0.879 |
| Ours | ✓ | **0.848** | **0.912** | 0.460 | **0.922** | **0.773** | **0.863** | 0.669 | 0.880 |

Table 1: **Quantitative results on ABC Koch et al. (2019) and ShapeNet Chang et al. (2015).**

Table 1 quantitatively compares our method with prior 3D CAD reconstruction and CNC-based approaches on the ABC Koch et al. (2019) and ShapeNet Chang et al. (2015) datasets. Our method achieves state-of-the-art IoU and F1 scores on both datasets, outperforming existing methods, with IoU/F1 of $0.848/0.912$ on ABC and $0.773/0.863$ on ShapeNet. While we surpass CNC-Net on CD and NC, our approach trails some CAD-specific models like CAPRI-Net, reflecting the difference between CNC-based carving and CAD's smooth primitive assembly, which is consistent with previous observations Yavatanoo et al. (2024). Most prior works are reported in their per sample fine-tuned setting to achieve manufacturing-grade fidelity. For CNC-Net and for our method, we additionally report a no-fine-tuning variant to assess zero-shot performance. In this no-fine-tuning setting, our method outperforms most of fine-tuned baselines on most metrics and shows stronger zero-shot generalization than CNC-Net. Furthermore, we present qualitative results in Figures 4a and 4b, where prior methods are visualized using marching cubes at a resolution of 256, while our results are shown through toolpath simulations shown in Figure 5a in the CIMCO CIMCO A/S G-code simulator. Unlike CAD-based methods that may miss localized features, our G-code simulation result demonstrates high fidelity and manufacturability, highlighting the practical effectiveness of our approach for real-world CNC applications.

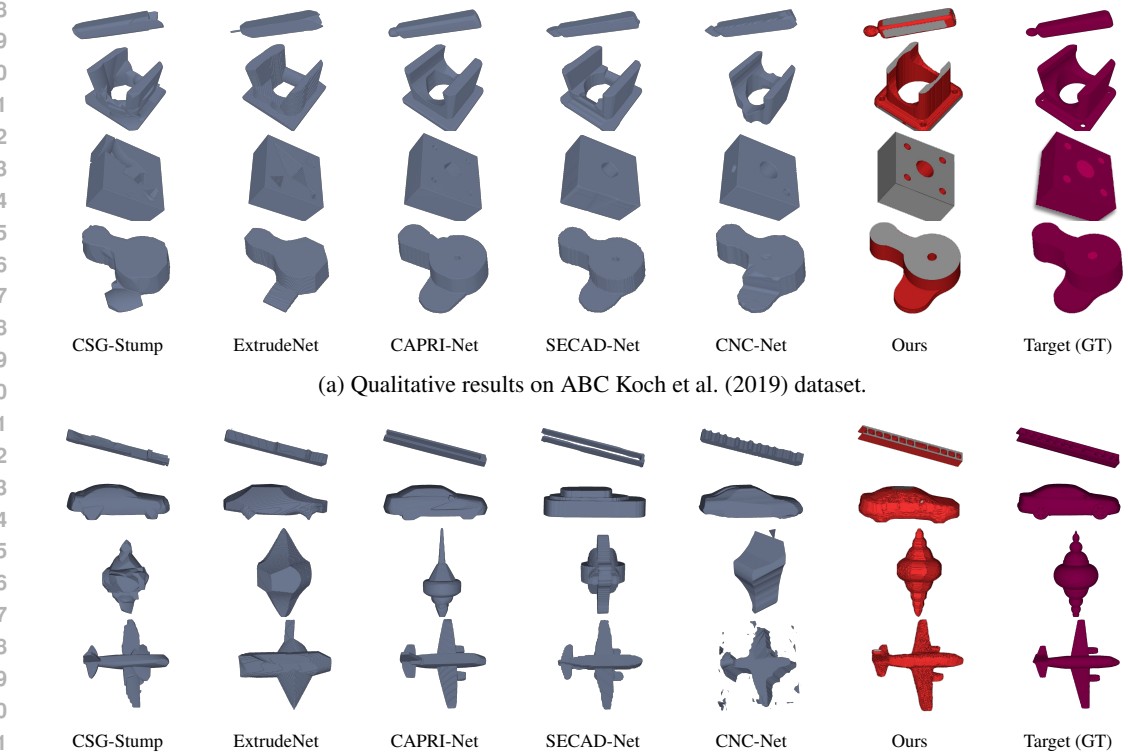

(a) Qualitative results on ABC Koch et al. (2019) dataset.

(b) Qualitative results on ShapeNet Chang et al. (2015) dataset.

Figure 4: **Qualitative comparisons.** Our results are produced using CIMCO G-code simulator. Red regions indicate the areas removed by cutting operations, while gray regions correspond to the untouched portions of the original stock material.

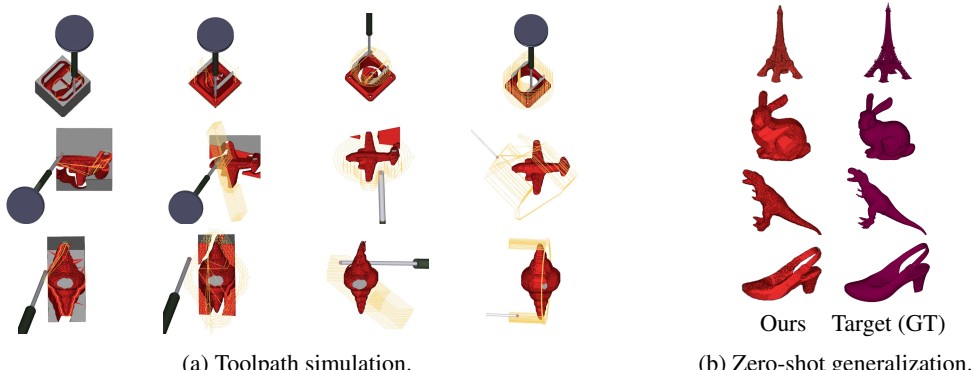

(a) Toolpath simulation.

(b) Zero-shot generalization.

Figure 5: **Visualizations of toolpath simulation (left) and zero-shot generalization (right).**

## 4.5 ABLATION STUDIES

### 4.5.1 EFFECT OF TOOL ORIENTATION SELECTION

To assess our tool orientation selection method, we compare three strategies: (1) six fixed principal-axis directions, (2) selecting multiple uniformly random orientations (averaged over several trials), and (3) our proposed approach. We evaluate each by the percentage of visible surface points across all selected views, using $16,384$ sampled points per object on various shapes from the ABC and ShapeNet datasets. As shown in Table 2, our method achieves greater surface coverage with fewer unseen points, using a set of five selected views, compared to fixed or randomly chosen orientations. This strategic optimization improves machining accessibility and efficiency.

| Dataset | 6-axes | Rand(5) | Rand(6) | Ours |
|---|---|---|---|---|
| **ABC** | 92.67% | 83.52% | 89.11% | **96.42%** |
| **ShapeNet** | 91.27% | 79.77% | 84.73% | **92.75%** |

Table 2: **Effect of tool orientation selection.**

| Method | IoU↑ | F1↑ | CD↓ | NC↑ | T↓ |
|---|---|---|---|---|---|
| Random | 0.781 | 0.863 | 0.905 | 0.521 | 533.52 |
| Fixed (Tool radius: 0.001m, Layer height: 1/512m) | **0.856** | **0.917** | **0.427** | **0.936** | 1634.001 |
| Fixed (Tool radius: 0.01m, Layer height: 1/64m) | 0.718 | 0.818 | 0.666 | 0.853 | **129.07** |
| Ours | 0.848 | 0.912 | 0.460 | 0.922 | 423.145 |

Table 3: **Effect of machining precisions.**

### 4.5.2 EFFECT OF MACHINING PRECISIONS

We evaluate the effect of selecting machining precision settings, specifically tool radius and layer height. As shown in Table 3, fixed setting reveal clear trade-offs: coarse configurations improve efficiency but reduce accuracy, fine configurations enhance precision but significantly increase machining time, and random selection leads to overall lower performance. In contrast, our method learns to adaptively choose these parameters through reinforcement learning, achieving the best balance between reconstruction quality and machining efficiency.

### 4.5.3 EFFECT OF TOOLPATH STRATEGIES

To analyze the impact of different toolpath strategies, we perform an ablation study by removing each strategy individually. Using 50 random samples from the ABC dataset, we evaluate our method's performance after excluding contour, zigzag, periphery, or spiral toolpaths. Table 4 shows that removing any single strategy degrades both reconstruction precision (IoU, F1, CD, NC) and machining efficiency (T), demonstrating each strategy's importance. Notably, removing contour causes the largest drop in reconstruction fidelity, despite only minor changes in machining time, due to its relatively short execution time compared to spiral, periphery, and zigzag paths.

| Excluded Toolpath | Metric | | | | |
|---|---|---|---|---|---|
| | IoU↑ | F1↑ | CD↓ | NC↑ | T↓ |
| **Spiral** | 0.797 | 0.876 | 0.615 | 0.911 | 508.132 |
| **Periphery** | 0.806 | 0.882 | 0.565 | 0.914 | 528.967 |
| **Zigzag** | 0.811 | 0.886 | 0.611 | 0.917 | 441.278 |
| **Contour** | 0.756 | 0.849 | 0.891 | 0.847 | 444.430 |
| **None (Ours)** | **0.848** | **0.912** | **0.460** | **0.922** | **423.145** |

Table 4: **Effect of toolpath strategies.**

| Excluded Reward | Metric | | | | |
|---|---|---|---|---|---|
| | IoU↑ | F1↑ | CD↓ | NC↑ | T↓ |
| $\mathcal{R}_s$ | 0.792 | 0.871 | 0.603 | 0.903 | **413.583** |
| $\mathcal{R}_{\mathrm{MRR}}$ | **0.851** | **0.914** | **0.456** | 0.921 | 774.776 |
| **None (Ours)** | 0.848 | 0.912 | 0.460 | **0.922** | 423.145 |

Table 5: **Effect of reward functions.**

### 4.5.4 EFFECT OF REWARD FUNCTIONS

We analyze the impact of each component in the total reward function (Equation 13) via ablation study, with $\mathcal{R}_f$ always included to guarantee overall shape reconstruction. As shown in Table 5, removing $\mathcal{R}_s$ significantly reduces shape accuracy (e.g., IoU), highlighting its role in preserving target geometry. Excluding $\mathcal{R}_{\mathrm{MRR}}$ leads to the highest execution time, confirming its importance for efficient machining. The full reward achieves the best balance of accuracy and efficiency, demonstrating that each component contributes to overall system performance.

### 4.5.5 ZERO-SHOT RESULTS ON OUT-OF- DOMAIN DATA

We assess the model's zero-shot generalization to out-of-domain 3D shapes unseen during training. Specifically, the model is first pretrained on the ShapeNet dataset and then tested on a variety of unseen, out-of-domain 3D objects to evaluate its generalization capability. As shown in Figure 5b, the method reconstructs high-fidelity shapes for these novel inputs without fine-tuning, demonstrating strong generalization despite distribution shifts.

## 5 CONCLUSION

We introduced Shape2Gcode, a reinforcement learning-based framework for G-code generation in CNC machining. Shape2Gcode optimizes toolpath strategies, and machining parameters without relying on intermediate CAD models or manual tuning. Experiments demonstrate that Shape2Gcode improves surface coverage, machining efficiency, and toolpath stability, while ablation studies validate the importance of each component. By integrating AI-driven optimization with CNC manufacturing, Shape2Gcode enhances automation, precision, and efficiency in modern machining workflows.

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

## A APPENDIX

### A.1 IMPLEMENTATION DETAILS

The reinforcement learning model was trained over 300 episodes on a multi-object dataset for 5-axis CNC toolpath optimization. The machining process was simulated on a $256^3$ voxel grid, with voxel resolution adaptively scaled to object size. The starting point was placed at a safe height, offset by $0.1$m from the top of the shape in the tool direction. Grid resolutions were set to $1/256$m for contour and $1/64$m for the remaining strategies. We used 16,384 sampled surface points to compute 1024-dimensional features with DGCNN. The DQN consisted of two fully connected layers with 128 units each. Training was performed using the Adam optimizer (lr $= 1 \times 10^{-3}$, $\gamma = 0.95$) and an $\epsilon$-greedy policy decaying from 1.0 to 0.01. Each reward term was weighted by $\lambda_s = 1$, $\lambda_f = 1$, and $\lambda_{\text{MRR}} = 0.1$. Experiments were conducted on an NVIDIA Quadro RTX 8000 GPU, and generated G-code was validated in CIMCO A/S for toolpath inspection and collision checking. jjik

### A.2 G-CODE SIMULATION

We simulate machining by carving a voxel stock along a polyline toolpath using a simple swept-volume approximation. The toolpath is a $T \times 4$ array of waypoints $(x, y, z, f)$, where the flag $f \in \{0, 1\}$ marks the *ending* waypoint of each segment as rapid (G0, $f{=}0$) or cutting (G1, $f{=}1$); only G1 segments remove material. Safe moves are encoded as G0 segments, and layer height is implicitly determined by the $z$ coordinates of successive G1 segments. A step-by-step simulation procedure is given in **Algorithm A1**.

---

**Algorithm A1:** SIMULATE TOOLPATH

**Input:** $stock\_vox, toolpath \in \mathbb{R}^{T \times 4}, tool\_radius, sweep\_step, grid\_min, voxel\_size, tool\_axis$
**Output:** updated $stock\_vox$
$erase\_vox \leftarrow \mathbf{0}$
**for** $i \leftarrow 0$ **to** $T - 2$ **do**
    $start \leftarrow toolpath[i, 0{:}3]; \quad end \leftarrow toolpath[i{+}1, 0{:}3]$
    **if** $toolpath[i{+}1, 3] == 0$ **then**
        ⌞ **continue**
    $L \leftarrow end - start; \quad$ **if** $\|L\| = 0$ **then**
        ⌞ **continue**
    $L_u \leftarrow L/\|L\|; \quad D \leftarrow L \times tool\_axis; \quad D_u \leftarrow D/\|D\|; \quad N_u \leftarrow tool\_axis/\|tool\_axis\|$
    $n \leftarrow \lfloor \|L\|/sweep\_step \rfloor$
    Sample $n$ tuples $(l, d, n', r, \theta, \phi)$ with $l \sim \mathcal{U}(0, 1), d \sim \mathcal{U}(-1, 1), n' \sim \mathcal{U}(-1, 1), r \sim \mathcal{U}(0, tool\_radius),$
    $\theta \sim \mathcal{U}(0, \pi), \phi \sim \mathcal{U}(0, 2\pi)$
    Build $P$:
        • $p_s = start + [r \sin\theta \cos\phi, \ r \sin\theta \sin\phi, \ r \cos\theta]$
        • $p_c = start + l \|L\| L_u + (r \cos\theta) D_u + (r \sin\theta) N_u$
        • $p_r = start + l \|L\| L_u + (d \, tool\_radius) D_u + (n' \, tool\_radius) N_u$
    Map $P$ to voxel indices: $v = \lfloor (P - grid\_min)/voxel\_size \rfloor; erase\_vox[v_x, v_y, v_z] \leftarrow 1$
$stock\_vox \leftarrow stock\_vox \odot (1 - erase\_vox)$
**return** $stock\_vox$

---

### A.3 MORE QUALITATIVE RESULTS

We extend our qualitative evaluation by providing additional compariosns between our method and existing 3D CAD reconstruction approaches, including CSG-Stump, ExtrudeNet, CAPRI-Net, SECAD-Net, and CNC-Net, on both the ABC and ShapeNet datasets. As shown in Figure A1, our method produces geometry that closely matches the ground-truth while remaining compatible with real world CNC manufacturing process. The visualizations are obtained via CIMCO G-code simulation, where red regions represent material removed during cutting, and gray regions indicate untouched portions of the original stock. These examples further highlight our model's ability to preserve both structural integrity and surface fidelity across diverse shapes, reinforcing its practical utility in real-world CNC applications.

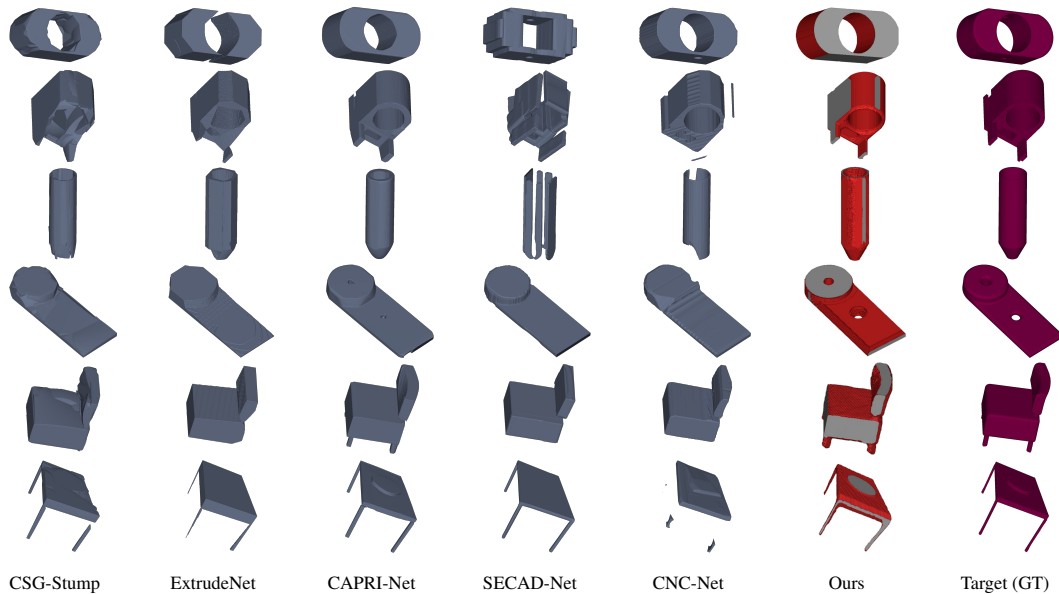

| CSG-Stump | ExtrudeNet | CAPRI-Net | SECAD-Net | CNC-Net | Ours | Target (GT) |

Figure A1: **Qualitative comparisons on ABC and ShapeNet dataset.** Our results are produced using CIMCO G-code simulator. Red regions indicate the areas removed by cutting operations, while gray regions correspond to the untouched portions of the original stock material.

## A.4 COMPARISON WITH COMMERCIAL CAM

We compare against MeshCAM, a representative commercial CAM tool, under two configurations. (1) *non-expert setting*: reasonable orientations with median tool radius and layer height. and (2) *expert setting*: a skilled machinist manually tunes orientation, tool size, and step size per toolpath strategy. As shown in Table A1, our method attains the best accuracy on all metrics, surpassing both MeshCAM settings. In machining time, our approach is far faster than the non-expert configuration and competitive with the expert setting (423.1 vs. 375.7). It is also efficient at runtime: ~5s for orientation selection, < 0.1ms for action selection, and ~10s for G-code translation, faster than MeshCAM.

Expert CAM typically relies on CAD metadata (e.g., features, tolerances). With only a triangulated mesh, experts must infer features and often choose conservative parameters. Our policy instead directly optimizes orientations and strategy from the mesh via fast simulation, exploring candidates over a few episodes and finding geometry-aware settings that are hard to hand-tune consistently. Overall, we outperform commercial CAM in accuracy while approaching expert-level efficiency without human intervention.

| Method | Accuracy & Machining Time | | | | | Runtime | | |
|---|---|---|---|---|---|---|---|---|
| | IoU↑ | F1↑ | CD↓ | NC↑ | T↓ | Orientation Sel. | Action Sel. | G-code Trans. |
| MeshCAM (Non-expert setting) | 0.783 | 0.854 | 1.474 | 0.868 | 1488.621 | – | – | 29.282 s |
| MeshCAM (Expert setting) | 0.833 | 0.904 | 0.555 | 0.910 | **375.708** | – | – | 18.759 s |
| Ours | **0.848** | **0.912** | **0.460** | **0.922** | 423.145 | 5081.98 ms | 0.03 ms | **10.103 s** |

Table A1: **Quantitative and runtime comparison on the ABC dataset.**

## A.5 ABLATIONS FOR EFFECTIVENESS OF PARAMETER SEARCH

To demonstrate the effectiveness of our parameter selection strategy, we compare our method against a baseline that adopts fixed parameters, specifically the largest tool radius (0.01m) and the lowest layer height (1/64m), from the action space. As illustrated in Figure A2, our method consistently generates high-fidelity reconstructions across a range of shapes, closely matching the target geometry. In contrast, the fixed-parameter baseline produces degraded or incomplete results, highlighting the

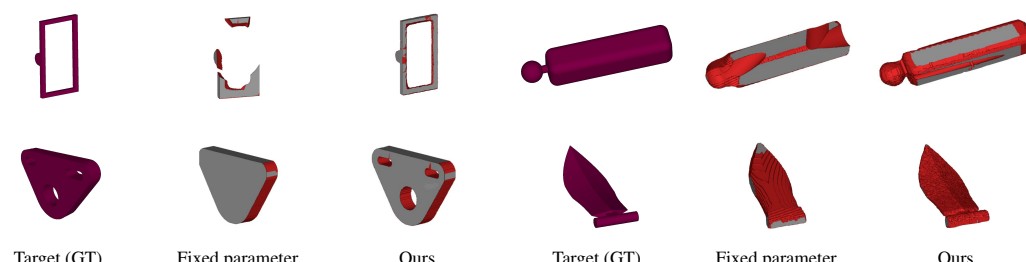

Figure A2: Comparison results across different examples showing the target object, results using fixed parameters with the largest tool radius (0.01m) and lowest layer height (1/64m), and our method.

importance of adaptive parameter selection. These results confirm that our approach successfully balances geometric accuracy and machining efficiency by selecting context-aware parameters.

## A.6 ABLATIONS FOR TOOL ORIENTATION SELECTION

| Method | IoU↑ | F1↑ | CD↓ | NC↑ |
|--------|------|-----|-----|-----|
| 6-axes | 0.792 | 0.876 | 0.711 | 0.881 |
| Rand(5) | 0.740 | 0.839 | 0.710 | 0.869 |
| Rand(6) | 0.751 | 0.846 | 0.700 | 0.873 |
| Ours | **0.848** | **0.912** | **0.460** | **0.922** |

Table A2: **Quantitative comparison of tool orientation strategies.**

| Method | Fine-tuning Time per Sample |
|--------|------------------------------|
| CSG-Stump | 60 min |
| ExtrudeNet | 30 min |
| CAPRI-Net | 3 min |
| SECAD-Net | 3 min |
| CNC-Net | 50 min |
| **Ours** | **10 min** |

Table A3: **Per-sample fine-tuning time comparison.**

We demonstrate the effectiveness of our proposed orientation selection strategy through comprehensive quantitative evaluations on ABC dataset. As shown in Table A2, our method consistently outperforms baseline approaches such as Random-5, Random-6, and 6-axis orientation selection across all mesh comparison metrics: Intersection over Union (IoU), surface fidelity (FI), Chamfer Distance (CD), and normal consistency (NC). Random-5 and Random-6 denote five and six randomly sampled orientations from a uniform distribution, while the 6-axis method uses the positive and negative directions of the Cartesian axes. Our method achieves the highest IoU, FI, and NC scores and the lowest CD values, highlighting the crucial role of our proposed orientation selection strategy.

## A.7 PER-SAMPLE FINE-TUNING TIME

Table A3 reports per-sample fine-tuning time. Our method requires **10 minutes** (five episodes) on a single NVIDIA Quadro RTX 8000 GPU, placing it between lightweight approaches such as CAPRI-Net and SECAD-Net (≈3 min) and optimization-heavy baselines including CSG-Stump (60 min), CNC-Net (50 min), and ExtrudeNet (30 min). Despite the per-instance adaptation, the minutes-scale budget keeps our procedure practical and competitive with prior work without introducing large additional computational overhead.

## A.8 MULTI-SEED ROBUSTNESS

To assess statistical reliability beyond a single random seed, we repeat each experiment with multiple random seeds and plot the mean with 95% confidence intervals (CIs) for each method and metric pair as shown in figure A3. Across datasets and metrics, our plots show tight CIs and stable rankings, indicating that the reported gains are not results of seed choice.

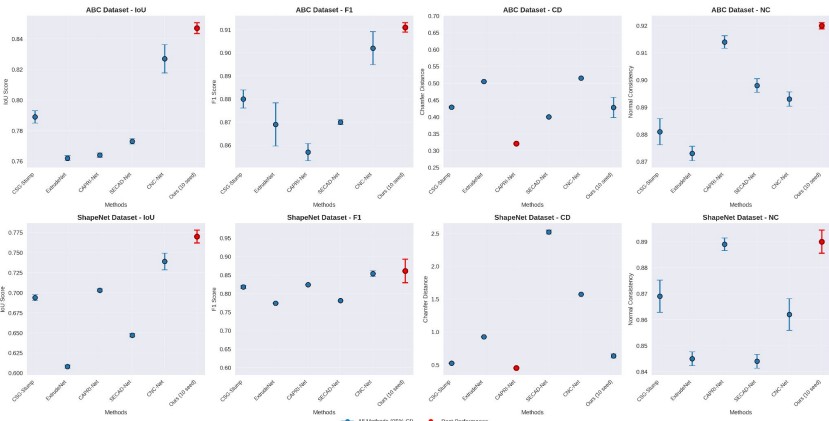

Figure A3: **Results over multiple random seeds.** Error bars indicate the 95% confidence interval of the mean. Red plots denote the best performance for each metric.

| Method | ShapeNet → OOD Dataset | | | |
|---|---|---|---|---|
| | IoU↑ | F1↑ | CD↓ | NC↑ |
| CNC-Net (w/ finetuning) | 0.630 | 0.765 | 1.234 | 0.780 |
| Ours (w/o finetuning) | **0.770** | **0.866** | **0.710** | **0.859** |

Table A4: **Quantitative results on OOD data.**

## A.9 QUANTITATIVE RESULTS ON OOD DATASET

We quantitatively evaluate ShapeNet→OOD transfer on a 20-shape benchmark (e.g., Stanford Bunny, Utah Teapot, Eiffel Tower). Both CNC-Net and our method are pretrained on ShapeNet. As CNC-Net is designed for per-sample fine-tuning, we allow CNC-Net to fine-tune on the OOD datasset, while our method uses performs without fine-tuning. As shown in Table A4, our approach outperforms CNC-Net across all metrics, demonstrating stronger OOD robustness with zero-shot adaptation.

