# OpenReview forum: "Shape2Gcode: Direct G-code Generation from 3D Shape Data for Automated Manufacturing"
_ICLR.cc/2026/Conference — ICLR 2026 Conference Withdrawn Submission_

### Official Review · Reviewer_PTqF · 2025-10-17

**Soundness:** 1
**Presentation:** 2
**Contribution:** 1
**Rating:** 2
**Confidence:** 3

**Summary:**

This paper investigates the potential for automating G-code generation, an important task which could greatly reduce the cost of manufacturing physical parts.  The paper aims to generate optimized G-Code directly, without a CAD or CAM system.   Reinforcement learning is used to select the toolpath strategy, along with tool radius and milling depth.   The accuracy with which the toolpath reconstructs the geometry is compared with a variety of CSG shape reconstruction methods and shown to perform better with both IoU and chamfer distance metrics.

**Strengths:**

This is an important task.   It’s the first time I have seen an attempt to generate G-Code directly from geometry.
The ability to go from a mesh model directly to optimized G-code would be extremely valuable.  Attempting to do this without any CAD or CAM system is an extremely challenging and audacious task.  To my knowledge this is the first attempt at solving this problem.

**Weaknesses:**

While the paper demonstrates that the generated toolpaths can be employed to cut the correct shape in simulation, the actual toolpaths generated may not be suitable to run on a real CNC mill.

I believe the initial contour strategy is performing a “full width cut”.   Cutting with the full width of the tool would usually be avoided when designing toolpaths.  Cuts of this kind have to be run extremely slowly or they will break the tool.   Most good roughing toolpath strategies try to control the tools engagement angle.  I recommend https://www.cnccookbook.com/ to the authors as a great source of information on this kind of consideration.

There is no discussion of feeds and speeds or cutting forces.  Optimizing a toolpath is very much focused on removing the requisite material in the fastest time while minimizing the cutting forces, avoiding tool breakage and maximizing tool life.  The authors may not have hand on experience operating CNC machine tools and clearly the paper was not focusing in this area, however omitting these very important considerations means that the reward functions used in the RL is solving a rather different problem to what is claimed in the abstract and introduction.

The radius of the tool is not considered when computing points on the surface which are accessible from a given cutting direction.

Action space is quite limited.  The majority of the toolpath generation is performed by something closer to traditional algorithms.  Most of the good results for IoU and chamfer distance come from this hand crafted CAM algorithm.

The work avoids computing leads and links and retracts the tool before each layer.  This is very inefficient.

**Questions:**

CNC-Net Yavatanoo et al. (2024) is mentioned in the introduction but is not referenced until the related work section.  I suggest adding this reference at the first mention (line 048)

> An angle selection module first determines valid machining orientations V = {vs} n s=1 to ensure full surface coverage.
> Each selected angle provides a different view of the object, ensuring that critical features are not occluded during machining

This doesn’t discuss multiple setups.   Using a high cost 5-axis machine tool you can change the tool axis, but still need to clamp the model somehow. Usually you want to use a low cost 3-axis CNC machine tool, so the tool axis is fixed and you need to halt the matching and get a person to set up the part in a new operation.  Needless to say, this takes a lot of operator time and is a major cost driver.

Line 147
- Using this learned features
+ Using these learned features

Line 178
> A surface point p is considered visible from a candidate orientation v if the cutting tool can reach p directly, without obstruction.
> Mathematically, visibility is determined by casting a ray from p along −v and checking for intersections with the object surface;
> if the ray hits another surface point before exiting, p is occluded in direction v.

This isn’t correct.  It doesn’t take account of the tool radius or the shape of the cutter.   Tool radius is important here when we have overhangs or narrow channels.   Also the radius of curvature of convex corners puts limits on the maximum radius of a ball nose tool.   Most prismatic parts are machined using an endmill.

Line 209
> At each layer l, the contour toolpath is first applied to separate the object’s main geometry from the surrounding material.
Sounds like you are applying a full width cut.  This would typically break the tool unless you are very careful.  The paper should contain warnings that toolpaths generated by these algorithms may not be suitable for running on real hardware.

> In each layer, the tool retracts to a starting position

This is really inefficient.  CAM software tries to avoid this.

Line 271
Remove repeated comma

---

### Official Review · Reviewer_nGnv · 2025-10-28

**Soundness:** 3
**Presentation:** 3
**Contribution:** 3
**Rating:** 6
**Confidence:** 3

**Summary:**

This paper proposes to automatically generate G-Code from CNC machines directly from a 3d model using Reinforcement Learning. Their method, Shape2Gcode, trains a Deep Q-Network (DQN) to pick the best tool size, cutting depth, and path strategy to carve out a shape for a set of greedily calculated tool angles. The authors test Shape2Gcode on the ABC and ShapeNet datasets. The approach creates more accurate results than previous CNC methods in both finetuning and direct test settings, while performing on par with existing CAD based approaches. Several ablations validate the effectiveness of the proposed action space and reward terms.

**Strengths:**

- The paper tackles a practical and important problem. Automating how G-code is generated for CNC machines could save a lot of time and specialized effort. Using RL to go straight from a 3D model to machine instructions is an interesting and promising idea.
- The approach itself is solid. Using a DQN makes sense for this problem, and I appreciate that the authors combined the deep RL setup with classical, non-learned elements like the module for selecting tool orientations and using standard toolpaths.
- The experimental results are overall convincing, especially for the provided qualitative examples. The authors compare their method against several recent approaches, and the ablation studies highlight the importance of different parts of the system. Testing both with and without finetuning is also appreciated, as this essentially shows that a trained model can be either used out-of-the-box, or can be further improved given some finetuning budget.
- The paper is easy to follow overall. The authors clearly explain their motivation and how their work improves upon previous methods. The figures are helpful in understanding the overall pipeline.

**Weaknesses:**

- The method seems to rely on a $256^3$ voxel grid, which might limit its ability to produce very fine or detailed parts. It would be helpful to discuss this aspect for high-precision manufacturing. For example, if this is the reason why it does not outperform CAD methods on surface-based metrics, clearly stating so would clarify the applications and limitations of the method.
- The "end-to-end" claim feels a bit strong, since some key parts of the process aren't learned. For instance, the tool orientation is chosen by a separate greedy algorithm, and the initial object outline seems to be carved out using a fixed procedure. It appears the RL agent is mostly learning how to efficiently clear away leftover material, rather than learning the entire carving process from scratch. Clarifying this scope would make the paper's claims more precise.
- Related to the previous point, its not clear where and if the presented method can fail and how difficult/sophisticated the RL part of the pipeline actually is. Judging by the ablations, the only real modes of failure are choosing a too-large tool radius and layer height. Its also not clear how and why the non-Contour toolpath strategies affect the shape metrics, since from the text, these should mostly be responsible for getting rid of excess material.
- There are a few typos and small grammatical mistakes throughout the paper that make it seem slightly rushed. I understand that these could easily be cleaned up in a revision, so this is a very minor weakness.

**Questions:**

1. The ablation shows that the proposed greedy orientation selection method finds more visible points. How much does this actually help the final machining quality? What happens to the IoU and other metrics after RL training on, e.g., the simpler "6-axes" orientation strategy or a more excessive sampling of, say, 10 orientations?
2. Table 3 compares to two extremes of coarse and fine settings for the machining precision. How does a more moderate, fixed setting, like a mid-range tool radius and layer height perform? It would be interesting to see how the method compares to a more reasonable hand-picked heuristic.
3. It would be really insightful to see a concrete example of the agent making non-trivial, intelligent decisions. Are there cases where certain toolpaths are significantly better than others, and why? When and how do layer height and tool width matter and are not immediately obvious from the geometry?
4. The multi-seed analysis in the appendix provides much-appreciated statistical significance to the results. Is there a reason why the experiments over multiple seeds are not included in the main paper?

---

### Official Review · Reviewer_g8o1 · 2025-10-29

**Soundness:** 2
**Presentation:** 2
**Contribution:** 2
**Rating:** 2
**Confidence:** 3

**Summary:**

In this paper, the authors propose a novel method for generating Gcode from 3D shape models that avoids the usual intermediary steps dependent on CAD tools. Using reinforcement learning, their method optimizes key machine parameters.

**Strengths:**

This paper tackles a practical problem of generating G-code. They provide a justified problem and a clear approach. They compare their method to relevant baselines on 2 datasets and show its benefits.

**Weaknesses:**

The paper writing can be improved. The main issue is the lack of citations. Citations to important works such as DQN are missing. In the introduction, the reference for CNC-Net is not included. *Marching Cubes* is another missed citation. In Section 3.2, the authors mention the surface voxelization approach and Laplacian smoothing, but no prior work is referenced.

It is not clear in the main part of the paper whether the authors considered multiple seeds. However, in Appendix A.8, they state that they did, but do not specify how many. Moreover, it is unclear why the values of the confidence intervals are not provided in the main results.

The dimensionality of the action space is also unclear. Specifically, in Equation 2, the authors define the action space as consisting of three parts, but the range of their values is not explained.

In Section 3.4.4 and Equation 13, the reward function is defined as a linear combination of multiple terms. However, additional ablation studies beyond those presented in Table 5 would be insightful.

In Table 1, the authors provide results for a fine-tuned version of their method. While fine-tuning is not necessary for the method to achieve good performance, additional explanation of the fine-tuning process would be beneficial.

Additional minor writing comments and typos are listed below:

- The caption for Figure 2 would benefit from additional explanation.
- Equations 6 and 8 appear identical; using different notation would help avoid confusion.
- Line 271 contains an extra comma.
- In line 284, the acronym *IoU* is introduced but defined later in Section 4.3.
- Line 606 contains what appears to be random text (“jjik”).

**Questions:**

Please see **Weaknesses**, especially the unclear points that are mentioned

---

### Official Review · Reviewer_gBGm · 2025-10-31

**Soundness:** 2
**Presentation:** 2
**Contribution:** 2
**Rating:** 2
**Confidence:** 3

**Summary:**

The authors propose Shape2Gcode framework that is using RL to optimize machining parameters for G-code generation in CNC machining. The approach is aimed at improving CNC manufacturing efficiency and enabling automated machining workflows. The results are showcased on ABC and ShapeNet datasets and compared to prior CAD reconstruction methods. Metrics show a marginal quantitative improvement.

**Strengths:**

- automation in manufacturing is a meaningful challenge with potential industrial applications
- ablations on toolpath strategies and rewards
- integration with a real G-code simulator for validation is commendable

**Weaknesses:**

- the RL formulation appears underexplored as the DQN acts as a discrete parameter selector not a true policy learning agent
- i'm unsure if laplacian smoothing is inflating metrics by reducing the voxel noise. The paper does not clarify whether the baselines received equivalent smoothing, and the reported IoU improvements (table 1) seem to fall within smoothing variance
- the baselines (CSG-Stump, CAPRI-Net) are CAD reconstruction methods optimized for parametric modeling and not machining (manufacturing), making the comparison somewhat misaligned
- optimal rotation matrices are mentioned in the abstract but are never mathematically defined.

**Questions:**

1. line 071 typo - "enusre" -> "ensure"
2. how is this approach different from using mastercam or Fusion360 in a headless manner to automate G-code generation using fixed/adaptive parameters?
3. how is tool collision or non-machinable geometry avoided when generating G-code from unseen shapes?
4. Does the method handle thin walls, threads, or multi-tool operations?
5. Can this framework currently prevent tool collisions with fixtures/clamps?
6. in Table A1, why does expert MeshCAM setting underperform the proposed method? this seems counterintuitive, given the expert knowledge.

---

### Note · Authors · 2025-11-12

**Comment:**

We have decided to withdraw this manuscript in order to incorporate additional experiments and improvements before resubmission. Thank you for you time and consideration.

**Withdrawal Confirmation:**

I have read and agree with the venue's withdrawal policy on behalf of myself and my co-authors.